# A Specific Interaction between Ionic Liquids’ Cations and Reichardt’s Dye

**DOI:** 10.3390/molecules27217205

**Published:** 2022-10-25

**Authors:** Angelica Mero, Luca Guglielmero, Lorenzo Guazzelli, Felicia D’Andrea, Andrea Mezzetta, Christian Silvio Pomelli

**Affiliations:** 1Department of Pharmacy, University of Pisa, Via Bonanno Pisano 33, 56126 Pisa, Italy; 2Scuola Normale Superiore, Piazza dei Cavalieri 7, 56126 Pisa, Italy

**Keywords:** ionic liquids, solvatochromism, molecular probes, Reichardt’s Dye, charge transfer

## Abstract

Solvatochromic probes are often used to understand solvation environments at the molecular scale. In the case of ionic liquids constituted by an anion and a cation, which are designed and paired in order to obtain a low melting point and other desirable physicochemical properties, these two indivisible components can interact in a very different way with the probe. This is the case with one of the most common probes: Reichardt’s Dye. In the cases where the positive charge of the cation is delocalized on an aromatic ring such as imidazolium, the antibonding orbitals of the positively charged aromatic system are very similar in nature and energy to the LUMO of Reichardt’s Dye. This leads to an interesting, specific cation-probe interaction that can be used to elucidate the nature of the ionic liquids’ cations. Parallel computational and experimental investigations have been conducted to elucidate the nature of this interaction with respect to the molecular structure of the cation.

## 1. Introduction

Solvatochromism, the dependence of the absorption wavelength of a dye with respect to the solvent in which it is solvated [1,2,3,4,5], is a tool that is widely used for the study of the nature of the solvation environment. Although it was developed for the study of bulk solvents, it was later extended to more structured contexts such as membranes, micelle, etc. [5].

When the ET_N_ scales had been conceived and the related solvatochromic probes, N-phenolic betaines [6], had been synthesized for the first time, ionic liquids (ILs) were an almost unknown scientific curiosity, far beyond being the vibrant and extended research topic of the last 25 years. The chemical structures of ILs’ ions are designed and paired to reduce the crystal lattice packing efficiency, thus lowering the melting temperature. Shape is dominant over chemical nature, and the only constraint is the chemical stability of each ion in the presence of the other. This has led to a variety of very different chemical structures, especially in cations. Some common IL ions are reported in Figure 1. 

With respect to pure molecular solvents, two charged species are present instead of a single neutral one. In a pure molecular solvent, the probe interacts with a single type of solvent with all its moieties. Conversely, in a pure ionic liquid, there are two charged species are present, different interaction with different parts of the zwitterionic probe are possible. The solvatochromism of binary mixtures of molecular solvents is well studied [1,7], but the case of ILs is different since the molar ratio of the two components is not free but is strictly constrained to 1:1 by an essential physical condition: electroneutrality. There are hints in the literature of the specific solvatochromic effects of anions and cations in ionic liquids [8,9,10,11,12,13,14,15,16].

Reichardt’s Dye (Figure 2) is most commonly used among the above-mentioned class of solvatochromic probes. As with all betaines, in its ground state, it is a zwitterion. This electrostatic structure interacts with ILs, leading to the well-structured organization of the cybotactic region of the probe [17]:The negatively charged phenate groups are at the center of a large pocket, and up to two cations interact with the strong electrostatic and hydrogen bond interactions within the phenate groups.The pyridinium charge is sterically hindered by the ortho- and para-phenyl groups. The formally positive nitrogen atom lies at the bottom of two symmetric narrow holes. Two anions can interact electrostatically on both sides of the probe, but the amount of this interaction is determined by their size [17,18]. Smaller ions such as halogenides are able to be nearest, leading to a stronger interaction.Other ions are placed in order to balance the electrostatic interactions.

The above-described solvation shell structure is schematized in Figure 2c. 

In a successive computational study of the same phenomena [18], we found a very specific interaction, not previously reported, between Reichardt’s dye and ionic liquid cations. It is with ILs where the formal positive charge is localized on a nitrogen-containing aromatic heterocycle. In this specific kind of excitation, the LUMO orbital of Reichardt’s Dye is involved in the electronic excitation process along with one or more of the π* orbitals of the imidazolium group. These orbitals, very similar in energy, mix, leading to this new dye-cation charge-transfer excitation. Experimental evidence was found in the UV-VIS spectrum of two of the computationally studied ILs [18]. With the intention of shedding some light on the establishment of this specific interaction between Reichardt’s dye and ionic liquid cations, in the present work, this effect was investigated in more detail and for a larger number of cations both from the experimental and the computational point of view. 

Computationally inexpensive continuum methods can describe, sometimes very accurately, several kinds of effects of molecular solvents [19,20,21]. In the case of ionic liquids, the presence of an anion and a cation, carrying neat charges, leads to more specific interactions, such as the one described in this paper, that require a description at the supermolecular level with the inclusion of one or more explicit ions in the ab initio calculations [22,23,24,25].

## 2. Experimental Part 

DFT calculations were performed using GPU-based Terachem Code version 1.94 [26,27]. All geometries were fully optimized (Appendix A). Dispersion was enclosed using the Grimme’s D3 model [28]. Calculations were performed on a Linux-based GPU server equipped with Asus RT 2080Ti GPU. The CAM-B3LYP/6-311++G(d,p) level of calculation was used.

NMR spectra were recorded with a Bruker Avance II (Bruker Italia S.r.l., Milano, Italy), operating at 400 (^1^H) and 100 MHz (^13^C). 

The experimental measurements of the interaction between Reichardt’s dye 30 and the different bis-(trifluoromethylsulfonyl)imide ILs were carried out by dissolving small amounts of the probe in ca. 500 µL of ILs in order to obtain absorbance values ranging between 0.15 and 1. The solutions were stirred until complete dissolution of the probe and then transferred to a cell quartz cuvette for the UV-Vis analyses. Before measurements the ILs were accurately dried under reduced pressure, and the water content was estimated by Karl Fischer titration, using an SI Analytics coulometer (Titroline 75 000 KFtrace). All the studied ILs showed a water content below 150 ppm. UV-Vis absorption spectra were recorded on a Cary 300 Agilent Spectrometer (frame aperture at 2 nm, lamp change at 350 nm).

Spectra deconvolutions were performed using MagicPlot Pro 2.9.3 Mac version. Initial guess was adjusted by the operator using the package GUI. Standard thresholds and algorithm were used.

1-Methylimidazole; 1-benzylimidazole; 1-butylimidazole; 1-bromobutane; 1,6-dibromohexane; o-picoline; m-picoline; and p-picoline were purchased from Alfa Aesar. Lithium bis-(trifluoromethylsulfonyl)imide and bis-(trifluoromethane)sulfonimide were purchased from IOLITEC. Trioctylmethylphosphonium methylcarbonate (methanolic solution) was purchased from Proionic. 1-Bromoethane; 2-(Bromomethyl)-naphthalene; pyridine; 4-methylmorfoline; and 2,6-Diphenyl-4-(2,4,6-triphenylpyridinium-1-yl) phenolate (Reichardt’s dye 30) were purchased from Merck.

All substances were used as received, without further purification, unless differently specified. 4-Methyl-2-pentanone was purchased from Alfa Aesar, and all other organic solvents were purchased from Sigma Aldrich. All organic solvents were used without further purifications.

### 2.1. Synthesis of Trioctylmethylphosphonium Bis-(Trifluoromethylsulfonyl)Imide ([TOMP]Tf_2_N) (***1***)

The title compound was prepared according to a procedure previously reported by our research group. To a commercial methanolic solution (36% *w*/*w*) of trioctylmethylphosphonium methylcarbonate, 1 equiv of bis-(trifluoromethylsulfonyl)imide was added dropwise at 0 °C. The resulting mixture was stirred at room temperature for 1 h and extracted with dichloromethane. The solvent was then removed under reduced pressure to afford the IL in quantitative yield. ^1^H NMR and ^13^C NMR data were in accordance with those reported in the literature [29].

### 2.2. Synthesis of N,N-Ethylmethylmorpholinium Bromide ([C_1_C_2_Mor]Br) (***12***)

[C_1_C_2_Mor]Br was synthesized according to the procedure reported in our previous work [8].

Briefly, 1-bromoethane was added dropwise over 1 h to an equimolar solution of 4-methylmorpholine in acetonitrile (ca. 0.5 mmol in 200 mL) while stirring vigorously. The mixture was refluxed for 8 h. The molten salt was decanted, washed three times with CH_2_Cl_2_ and accurately dried under low pressure. The product was obtained as a white solid with a yield of 95%, and it was characterized by NMR spectroscopy. The obtained spectra were in agreement with the published data [8].

### 2.3. Synthesis of N-Butylpyridinium Bromide ([C_4_Pyr]Br) (***13***)

[C_4_Pyr]Br was synthesized by adding 1-bromobutane to an equimolar solution of pyridine in MeCN (70 mmol in 20 mL). The mixture was stirred at reflux for 18 h. Diethylether was added to the final mixture to precipitate the desired product and eliminate unreacted 1-bromobutane. Solid precipitate was filtered under vacuum, washed with diethylether and dried in vacuum, to afford a white solid product (yield of 94%). The synthesized compound was characterized by NMR spectroscopy, and the obtained spectra were in agreement with the data reported in the literature [30].

### 2.4. Synthesis of 1-Methyl-3-(Naphthalen-2-Ylmethyl)-1H-Imidazolium Bromide ([Naphc_1_im]Br) (***14***)

15 mmol (1.05 eq.) of 1-methylimidazolium diluted in 5 mL of MeCN were added to a solution of 2-(bromomethyl)naphthalene (1 eq.) in 5 mL MeCN. The mixture was reacted for 24 h at 70 °C. The crude reaction product was extracted with diethylether, and the desired product was obtained as viscous yellow oil after solvent removal under reduced pressure. ^1^H-NMR and ^13^C-NMR data were in accordance with those reported in the literature [31].

### 2.5. Synthesis of N-Butylpicolinium Bromides ([o-C_4_Pic]Br (15); [M-C_4_Pic]Br (16); [P-C_4_Pic]Br (***17***))

Picolinium bromides were synthesized by adding 1-bromobutane to an equimolar solution of the appropriate picoline (*o*-picoline, *m*-picoline or *p*-picoline) in THF (65 mmol in 20 ml). The mixture was stirred at reflux until the precipitation of the desired products occurred (24 h for [m-C_4_Pic]Br and [p-C_4_Pic]Br; 48 h for [o-C_4_Pic]Br). The [p-C_4_Pic]Br and [o-C_4_Pic]Br were obtained as white solids with yields of 91% and 72%, while the [m-C_4_Pic]Br was precipitated as transparent liquid (yield of 87%). All the synthesized compounds were washed three times with THF, accurately dried under vacuum and characterized using NMR spectroscopy, and the obtained spectra were in agreement with the data reported in the literature [30,32,33].

### 2.6. Synthesis of 1-Butyl-3-Methylimidazolium Bromide ([C_4_C_1_Im]Br) (***18***)

[C_4_C_1_Im]Br was obtained following a general procedure that was previously reported [34]. To a solution of 1-methylimidazole (1 equiv, 45 mmol) in THF 20 mL, 1-bromobutane (1.1 equiv) was added dropwise, and the resulting solution was stirred at reflux temperature for 24 h. The reaction solvent was separated, and the crude reaction product was washed with ether and ethyl acetate. The remaining solvent was removed under reduced pressure, affording a colorless oil with a 95% yield. ^1^H-NMR and ^13^C-NMR data were in accordance with those reported in the literature [35].

### 2.7. Synthesis of 1,3-Bis-(Phenylmethyl)Imidazolium Bromide ([(Bz)_2_Im]Br) (***19***)

20 mmol (1.05 eq.) of benzylbromide dissolved in 5 mL MeCN was added dropwise to a solution of 1-benzylmethylimidazole (1 eq.) in 5 mL MeCN. The mixture was reacted for 24 h at 70 °C. The crude reaction product was extracted with diethylether, and the desired product was obtained as viscous yellow oil after solvent removal under reduced pressure. ^1^H-NMR and ^13^C-NMR data were in accordance with those reported in the literature [36].

### 2.8. General Procedure for The Synthesis of 3,3′-(Hexane-1,6-Diyl)Bis-(1-Alkylimidazolium) Bromides (***20***–***21***) 

Compounds **20**–**21** were obtained following the general procedure previously reported [37]. 1,6-Dibrohexane (58 mmol, 1 equiv) and 10 mL of 4-methyl-2-pentanone (MIBK) were added in a three-necked flask. A solution of 1-alkylimidazole (122 mmol, 2.1 equiv) in MIBK (10 mL) was added dropwise under magnetic stirring at 0 °C. Another 10 mL of MIBK were added, and the solution was stirred for 15 min. The reaction mixture was then heated up and stirred at 80 °C for 12 h. The obtained solid precipitates were filtrated under vacuum, washed with MIBK (3 × 5 mL), and then, washed with EtOAc (3 × 5 mL) and dried in vacuo to afford white solids with excellent yields. ^1^H and ^13^C NMR data of compounds **10**–**11** were found to be in good agreement with those reported in the literature [37].

### 2.9. General Procedure for The Preparation of Bis-(Trifluoromethylsulfonyl)Imide Ils ***2***–***11***

Monocationic and dicationic bis-(trifluoromethylsulfonyl)imide ILs were obtained with two-step procedures reported in previous works [34,38] through a metathesis reaction from bromide precursors **12**–**21**. 

10 mmol of bromide salts (1 eq.) and lithium bis-(trifluoromethylsulfonyl)imide (1.05 eq.) were stirred at room temperature in water (5 mL) for 1 h. To the resulting biphasic system, dichloromethane (5 mL) was added, and the organic phase was extracted and washed with water until the silver assay on the washing waters resulted negative. The title compound was obtained, through drying under reduced pressure, as colorless or yellowish oils with about an 80% yield for dicationic systems and above 90% for monocationic ILs. ^1^H and ^13^C NMR data were found to be in good agreement with those reported in the literature [8,29,39,40,41,42,43,44]

## 3. Results and Discussion

Although the amount of available computational resources is still increasing over time, accurate calculations on clusters, such as the ones depicted in Figure 2c, are still computationally expensive, especially when we deal with nonground-state properties such as excitation energies.

Herein, we limited the model system to a complex of a single cation with Reichardt’s dye. This minimalistic choice, removing all other effects, allowed us to better focus on the effect of the cation on the electronic transitions. The choice of the Tf_2_N anion for the experimental measurements was functional to the above-defined model system. Indeed, the positively charged center of the dye was sterically hindered, which caused the large anion to be far away, with little influence on the solvatochromic effect [19]. 

All the model system’s geometries were optimized at the CAM-B3LYP/6-311++G(d,p) level of theory. D3 dispersive correction was included [28]. On the optimized geometries, a TD-DFT calculation on the first [26] excited states was performed. 

In order to depict how the chemical nature, the geometry and the electronic structure (strictly related quantities) of the cation influenced the effect under analysis, eleven different cations with different structures, reported in Figure 3, were chosen. 

Two cations without π delocalized systems were chosen to act as a blank test: the sterically hindered trioctylmethylphosphonium (TOMP) and (*N*,*N*)-ethylmethylmorpholinium (C_1_C_2_Mor). In the first cation, the positive charge was localized on the P atom and locked by the surrounding sidechains, while in the latter, the positively charged center was more sterically accessible. 

The first series of aromatic cations was based on the pyridinium nucleus *N*-butylpiridinium (C_4_Pyr) and the three geometrical isomers of *N*-butylpicolinium (o-C_4_Pic, m-C_4_Pic, p-C_4_Pic). The picolinium series allowed for the analysis of the effect of steric hindrance on the positively charged center. 

The second and last series of aromatic cations were based on the classic imidazolium ring with different sidechains: the very common 1-butyl-3-methylimidazolium (C_4_C_1_Im), 1,3-dibenzylimidazolium (Bz)_2_Im and two structures with two imidazolium nuclei separated by a C6 spacer (C_1_Im-C_6_-C_1_Im and C_4_Im-C_6_-C_4_Im). As mentioned previously, all the cations were paired with the Tf_2_N anion.

The geometrical arrangement of Reichardt’s dye and the cations was driven mainly by electrostatics: the positively charged center of the cation approached the phenolate moiety of the dye as much as possible. A second component of this interaction was the hydrogen bond between the acidic hydrogen atoms in the α position with respect to the cationic center. Consider Table 1 and the geometrical quantities defined in Figure 4. 

As can be seen from the values of R_C1-O_ and R_C2-H_, the internal geometry of the two fragments was not significantly affected by the cation-dye interaction, and all the values lay in a range of 0.05Å.

The distance R_OH_ is the first index of the intensity of the cation-dye interaction. It is strictly related to the accessibility of the cation atoms to the dye. 

A brief inspection of these values shows that the aliphatic ones had the larger values, especially for the sterically hindered TOMP cation; that the series with the pyridinic nucleus was consistent with the presence and/or the position of the other substituents on the ring; and that the imidazolium series showed the smallest values (the hydrogen was directly on the cationic ring). Variations in R_C2-H_ were small, as was usual for C-H distances. There were small variations in R_C1-O_ values also. The smaller the R_O-H_ value was, the larger the R_C1-O_ one was. This was coherent with the fact that a stronger interaction with the cation decreases the C1-O bond order. The angles were sensible with the steric features of the cation and are discussed in Figure 5.

The HOMO ➝ LUMO electronic transition of Reichardt’s dye was used to define the ET(30) solvatochromic scale [1]:(1)$$\mathrm{ET}\left(30\right)\;\left(\frac{\mathrm{Kcal}}{\mathrm{mol}}\right)=\frac{28591}{\lambda_{\max}}\left(\mathrm{nm}\right)$$
where $\lambda_{max}$ corresponds to the maximum absorbance. The normalized scale was also widely used:(2)$$E_{N}^{T}=\frac{\mathrm{ET}\left(30\right)-30.7}{32.4}$$
where 30.7 is the value of ET(30) of water and 32.7 is the difference between the ET(30) of TMS and water [1]. Since TMS and water showed, at the time of the scale definition, the largest and the smallest values of ET(30), the $E_{N}^{T}$ values were intended to lie between 0 and 1. 

After the scale was defined, some solvents, including some ionic liquids [8], displayed values outside this range [1]. 

An analysis of the molecular orbitals calculated at the DFT level is reported in Figure 6. With respect to the isolated dye (a condition that cannot be achieved experimentally), the HOMO and LUMO orbitals were modified, while the transfer charge density on the cation remained unchanged. Conversely, in the cation-related transition there was a significative transfer of electronic density to the cation.

The numerical data about these two transitions are reported in Table 2. For the two aliphatic cations, there was either no charge-transfer transition (TOMP), or it was very weak (C_1_C_2_Mor), as can be easily figured out from their molecular structure.

For the aromatic systems, while transition one was a “classical” HOMO ➝ LUMO transition, transition two involved higher-energy virtual orbitals. As reported in Figure 6, LUMO + 2 orbitals involved the π* orbitals of the charged aromatic center of the cation mixed with the analogous orbitals of the dye. The LUMO + 1 orbitals involved in this kind of transition were also π* orbitals but were localized on the dye only. Higher-energy orbitals were also involved in the transitions, as discussed later. Considering that there were several aromatic systems with similar electronic structures in the dye (lateral phenyl groups) and, in the dicationic case, also in the cation, there were several virtual orbitals very close in energy.

For the pyridinium series (C_4_Pyr, o-C_4_Pic, m-C_4_Pic, p-C_4_Pic), represented in Figure 5 and Table 1, the position of the methyl substituent on the ring influenced both the distance and the orientation of the cation with respect to the phenolic oxygen pocket: the distance C2-H in the unsubstituted cation C_4_Pyr was 0.5Å less than in o-C_4_Pic, which was the more hindered structure of the series.

This was reflected on the second electronic transition parameters: the closer the sidechain was to the charged center, the less intense the effect was.

The imidazolium series presented a common charged ring nucleus and different sidechains. As already described in the caption of Figure 5, there was not enough space for a simultaneous interaction of both charged rings with the phenolate; thus, the second ring acted as a sidechain. The geometrical arrangement of these systems was very similar. Only the dicationic systems C_1_Im-C_6_-C_1_Im and C_4_Im-C_6_-C_4_Im presented a R_O-H_ distance that was about 0.05 Å greater with respect to the other systems.

Systems C_4_C_1_Im, NaphC_1_Im and (Bz)_2_Im (with neutral sidechains) showed very similar secondary transitions. Although the geometrical arrangement of the sidechain and an inspection of the first elements of the virtual orbitals suggested possible π- π stacking with the lateral phenyl groups of the dye, this did not lead to a remarkable effect. Sidechains were involved in the case of the dicationic C_1_Im-C_6_-C_1_Im and C_1_Im-C_6_-C_1_Im systems where the main component of the transition’s excited state was LUMO + 1 (very similar to LUMO), but there was a significative contribution (about 40%) from the π* orbitals of the second charged ring that corresponded to LUMO + 4 and LUMO + 5 orbitals.

Moving to experimental spectra (Figure 7), some affinities at the qualitative level in the computational data were noticeable. Given the minimal choice of the model system, it was not possible to have a direct comparison between the two sets of computational and experimental data at the numeric/quantitative level.

The spectra pattern was expected in these kinds of systems and was reported elsewhere [18]: the cation-related transition peak was not isolated but superimposed on the big peak at a low wavelength. Therefore, the spectra were deconvoluted using a linear function and three gaussians using the MagicPlot Pro package. The deconvolution results are reported in Table 3. Only the TOMP spectrum did not present the above-mentioned shoulder. In this case, the deconvolution procedure did not converge with the three gaussians, while, after the elimination of one of the functions, it converged. This was qualitatively coherent with the computational results.

Disregarding all the surrounding solvent ions except one cation led to a difference of 50–100 nm in the position of the cation-related transition. Experimental values were all at a higher wavelength/lower energy which suggests a possible compensation effect due to neighboring anions. There were some hints that can confirm this hypothesis. The first is that, while computed values for mono-charged imidazolium series were very close, the experimental ones were more widespread: the size of the cation matters. The second one is that both experimental and computational data presented the same trend in the pyridinium series where the size of the cations was similar.

Last but not least, electrostatic-based intuition may suggest that the model system may be particularly inadequate in the case of dications. Unexpectedly, their experimental and computational values were the closest of all the studied systems. The charge-balancing anions were probably kept far from the dye chromophore, and their effect on the transition was reduced.

## 4. Conclusions

The computational and experimental data presented in this paper showed that the secondary effect exists at both the experimental and computational level. The roughness of the computational model was double-faced: the two series of values were very different, but this difference allow us to speculate on the influence of the size and nature of the cation on this kind of phenomenon and on cybotaxis of the negative extremity of the dye chromophore.

The existence of this effect introduces a possible new use for Reichardt’s dye in these kinds of systems.

The investigation can be extended by a more realistic model system, probably by using a QM/MM scheme, allowing it to further different kinds of ILs and, perhaps, to the emerging field of deep eutectic solvents.

## Figures and Tables

**Figure 1 molecules-27-07205-f001:**
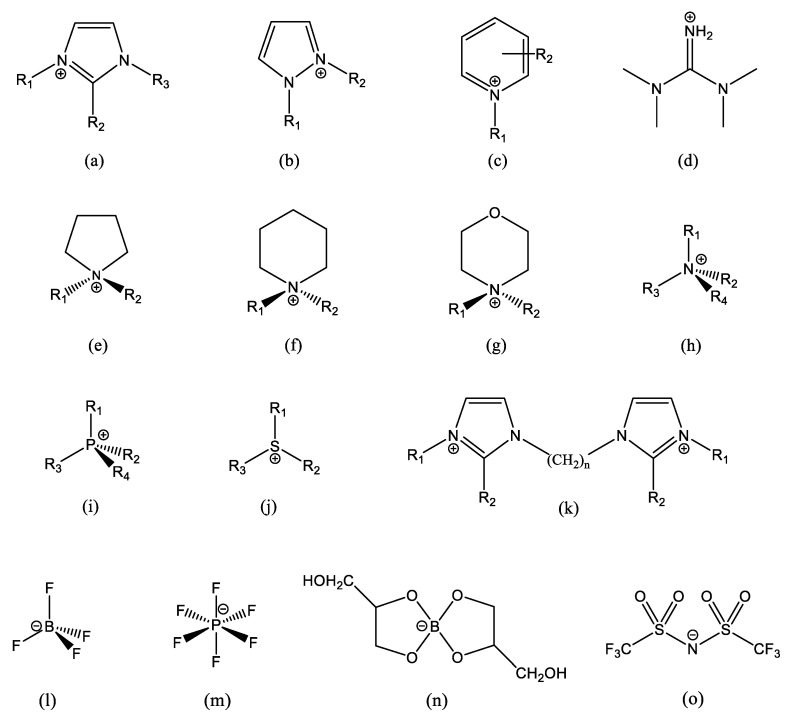
Some common IL ions, and some typical structures of IL cations and anions: (**a**) alkylimidazolium, often R_2_ = H, R_3_ = CH_3_ and R_1_ is a longer chain; (**b**) alkylpyrazolium; (**c**) alkylpyridinium; (**d**) tetramethylguanidinium; (**e**) alkylpyrrolidinium; (**f**) alkylpiperidinium; (**g**) alkylmorpholinium; (**h**) tetraalkylammonium; (**i**) tetraalkylphosphonium; (**j**) trialkylsolphonium; (**k**) an example of imidazolium-based dication; (**l**) tetrafluoroborate; (**m**) hexaflurophosphate; (**n**) glyceroborate, an example of an organic anion; and (**o**) bis-(trifluoromethane)sulfonimide, which has the short name bistriflimide and acronyms Tf_2_N and TFSI (a very common anion). In (**b**,**e**–**g**), R_1_ and R_2_ are different lengths. In (**h–j**), three (or two) sidechains are generally the same, and the 4th (or 3rd) is a very different length. Small organic/inorganic ions, such as Cl^−^, Br^−^, RCOO^−^, ^−^OOC(CH_2_)_n_COO^−^, RSO_3_^−^, etc., are also common.

**Figure 2 molecules-27-07205-f002:**
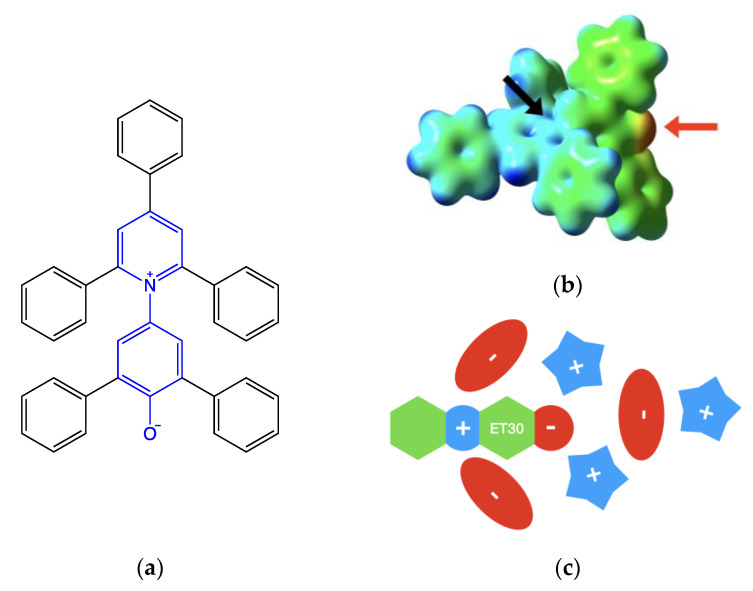
(**a**) Reichardt’s dye structure. The betaine core (in blue) presents five phenyl substituents. Steric hindrance does not allow the planarity of the molecule. (**b**) Isodensity map (threshold vale 0.01 a.u.) is mapped with the molecular electrostatic potential. False color scale goes from blue (electrophilic) to red (nucleophilic) while green (weak electrophilic) and yellow (weak nucleophilic) show regions with intermediate values. On the right is the phenate moiety (indicated by the red arrow). At the center is one of the sides of the pyridinium pocket (indicated by the black arrow). (**c**) Schematic representation is of the first solvation shell of Reichardt’s dye in ionic liquid.

**Figure 3 molecules-27-07205-f003:**
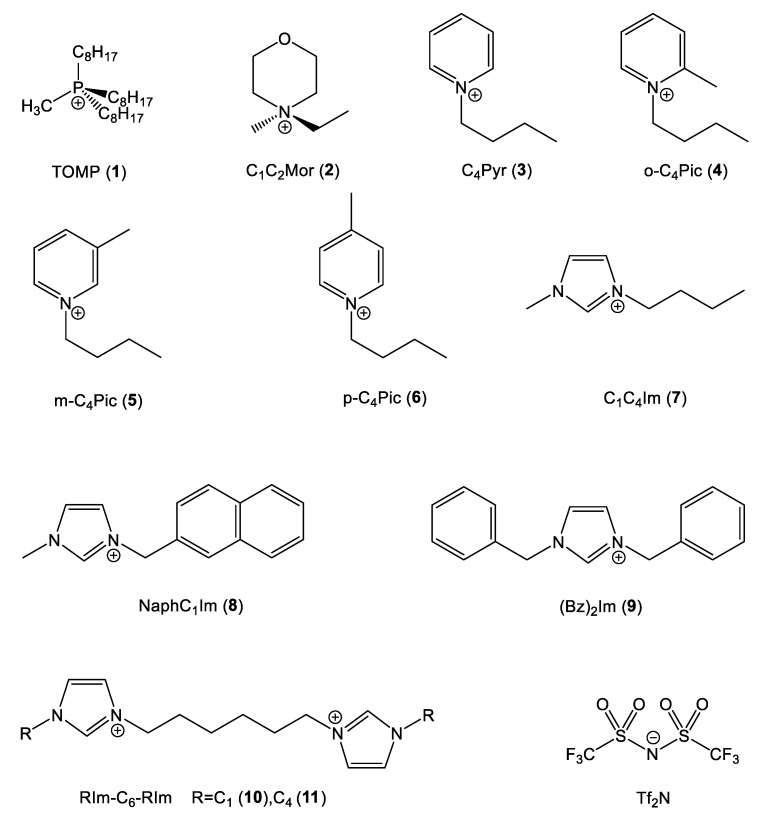
Cation and anion structures.

**Figure 4 molecules-27-07205-f004:**
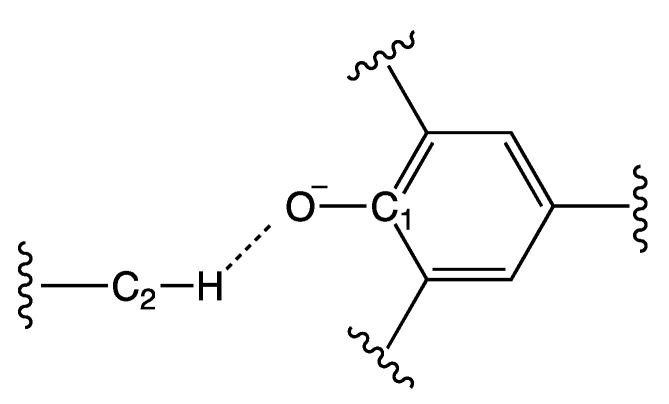
Definition of the labels used in Table 1. C1 represents the carbon atom of Reichardt’s Dye that is bonded to the oxygen O; C2 represents the carbon atom in α position with respect to the positive center nearest to the dye; and H represents the hydrogen atom bound to C2 nearest to the dye.

**Figure 5 molecules-27-07205-f005:**
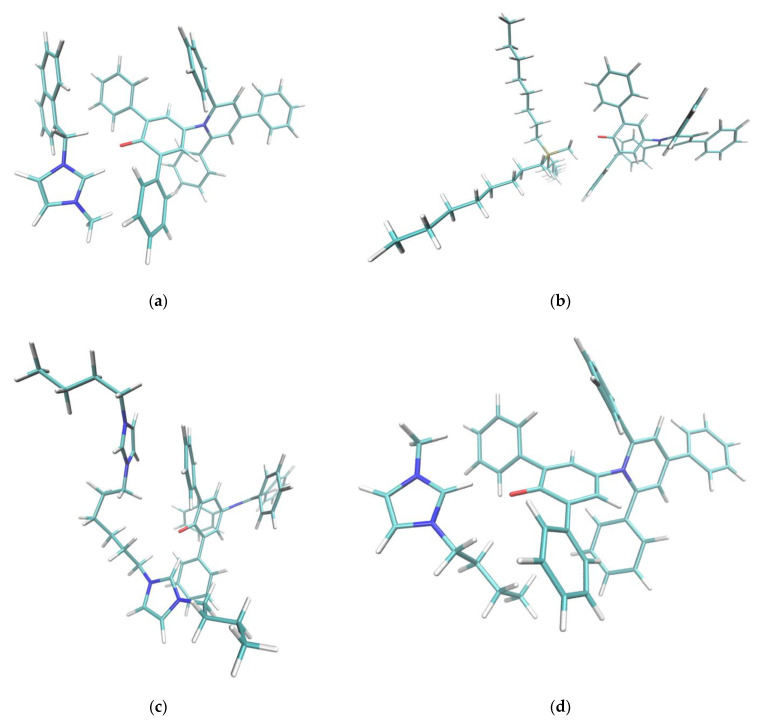
Cation-dye spatial arrangements for some of the cations studied in this paper. (**a**) NaphC_1_Im, (**b**) TOMP, (**c**) C_4_C_1_Im, (**d**) C_4_Im-C_6_-C_4_Im, (**e**) o-C_4_Pic, (**f**) m-C_4_Pic, (**g**) p-C_4_Pic. In the case of the sterically hindered TOMP cation (**b**), the size and the absence of orbital of the π kind did not allow the specific interaction described in this paper. Flat imidazolium- or pyridinium-based aromatic compounds (**a**,**c**–**g**) could approach the oxygen moiety at the center of the Reichardt’s Dye pocket more efficiently. In the case of the three picoline isomers (**e**–**g**), when the methyl substituent was nearby the positively charged center (ortho and meta), it forced a different interaction, weakening the effect. The sidechains that presented an aromatic π system (**a**,**d**) stacked efficiently on one of the two lateral phenylic groups of the dye molecule. In the case of the dicationic system (**d**), steric hindrance did not allow for the simultaneous interaction of both the positive centers with the phenolic oxygen: one interacted, and the other acted as sidechain. Small sidechains (**c**,**g**) on imidazolium did not affect the position and orientation of the cation with respect to the dye molecule.

**Figure 6 molecules-27-07205-f006:**
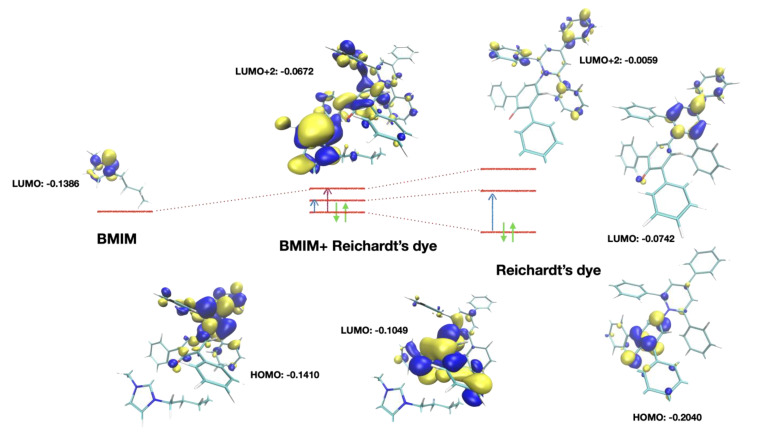
Orbitals involved in the transitions. Right: free Reichardt’s dye. The HOMO ➝ LUMO transition (blue arrow) involved an internal charge transfer. In the dye-cation cluster, the LUMO + 2 orbital mixed with the LUMO of the cation, leading to shared orbitals where there was dye-cation charge transfer (purple arrow). The actual transition also involved LUMO + 3 and LUMO + 4 orbitals, similar to LUMO + 2, but they were not reported here.

**Figure 7 molecules-27-07205-f007:**
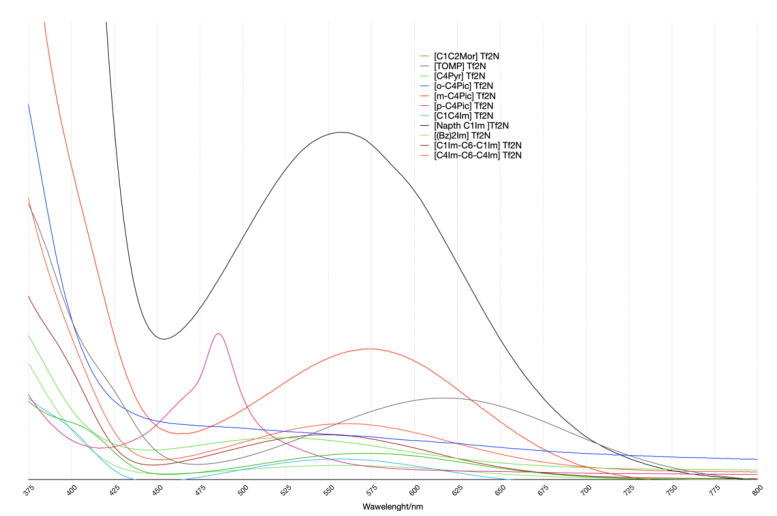
UV-VIS spectra. All Spectra were renormalized with respect to the area of the peak related to the π ⟶π^*^ transition (its maximum is at λ_1_ wavelength of Table 3). This was for a better comparison of the several systems studied. Thus, the vertical axis corresponds to arbitrary units.

**Table 1 molecules-27-07205-t001:** Some geometric quantities related to cation-Reichardt’s dye 1:1 complexes.

IL	R_O-H_/Å	R_C1-O_/Å	R_C2-H_/Å	∡_C1-O-H_/˚	∡_C2-H-O_/˚
none	-	1.293	-	-	-
[TOMP]Tf_2_N (**1**)	3.635	1.253	1.089	96.7	167.9
[C_1_C_2_Mor]Tf_2_N (**2**)	2.503	1.254	1.086	80.4	85.8
[C_4_Pyr]Tf_2_N (**3**)	1.793	1.268	1.097	110.8	139.7
[o-C_4_Pic]Tf_2_N (**4**)	2.310	1.261	1.083	103.0	133.3
[m-C_4_Pic]Tf_2_N (**5**)	2.265	1.263	1.087	116.4	133.1
[p-C_4_Pic]Tf_2_N (**6**)	2.061	1.267	1.091	134.3	144.1
[C_4_C_1_Im]Tf_2_N (**7**)	1.729	1.268	1.099	146.2	146.9
[NaphC_1_Im]Tf_2_N (**8**)	1.732	1.271	1.099	146.1	140.2
[(Bz)_2_Im]Tf_2_N (**9**)	1.770	1.271	1.099	145.6	138.2
[C_1_Im-C_6_-C_1_Im]Tf_2_N (**10**)	1.821	1.277	1.096	153.1	125.5
[C_4_Im-C_6_-C_4_Im]Tf_2_N (**11**)	1.812	1.278	1.098	153.4	125.3

**Table 2 molecules-27-07205-t002:** Data on TDDFT excitations. Transition one was the usual HOMO ➝ LUMO transition related to the standard charge-transfer effect of the solvatochromic probe. Transition two was a transition between HOMO and a mix of virtual orbitals where the main component was the LUMO + N one.

IL	λ_1_/nm	osc_1_	λ_2_/nm	osc_2_	N
none	659.8	0.4076	-	-	-
[TOMP]Tf_2_N (**1**)	589.4	0.2343	-	-	-
[C_1_C_2_Mor]Tf_2_N (**2**)	563.5	0.2008	289.8	0.0001	+ 2
[C_4_Pyr]Tf_2_N (**3**)	505.5	0.1669	416.2	0.0031	+ 2
[o-C_4_Pic]Tf_2_N (**4**)	527.6	0.1916	446.3	0.0006	+ 2
[m-C_4_Pic]Tf_2_N (**5**)	522.9	0.2200	432.7	0.0142	+ 2
[p-C_4_Pic]Tf_2_N (**6**)	518.3	0.1769	429.7	0.0161	+ 2
[C_4_C_1_Im]Tf_2_N (**7**)	502.0	0.1823	309.1	0.1933	+ 2
[NaphC_1_Im]Tf_2_N (**8**)	502.7	0.1964	309.9	0.1882	+ 2
[(Bz)_2_Im]Tf_2_N (**9**)	506.3	0.1873	308.7	0.1890	+ 2
[C_1_Im-C_6_-C_1_Im]Tf_2_N (**10**)	429.4	0.2343	324.8	0.1391	+ 1
[C_4_Im-C_6_-C_4_Im]Tf_2_N (**11**)	424.2	0.1619	327.2	0.1214	+ 1

**Table 3 molecules-27-07205-t003:** Deconvolution fitting of UV-VIS spectra. H is the half-width at half-height. The areas A were normalized with respect to the area of transition 1.

IL	λ_1_/nm	H_1_/nm	λ_2_/nm	A_2_/A_1_	H_2_/nm	λ_3_/nm	A_3_/A_1_	H_3_/nm	a	b
[TOMP]Tf_2_N (**1**)	598.9	59.9	-	-	-	151.3	152.4	105.1	0.0	0.3
[C_1_C_2_Mor]Tf_2_N (**2**)	569.9	75.6	406.9	27.6	20.4	314.3	441.4	56.5	0.0	0.1
[C_4_Pyr]Tf_2_N (**3**)	535.6	68.1	322.0	2.60	12.4	85.3	409.7	119.3	0.0	0.0
[o-C_4_Pic]Tf_2_N (**4**)	390.1	16.8	371.7	0.75	10.2	321.5	18.8	26.9	0.0	0.2
[m-C_4_Pic]Tf_2_N (**5**)	582.0	55.8	383.9	1.54	24.3	315.9	16.1	28.35	0.0	0.2
[p-C_4_Pic]Tf_2_N (**6**)	483.7	21.2	362.9	0.56	17.5	309.4	11.5	22.3	0.0	0.3
[C_4_C_1_Im]Tf_2_N (**7**)	554.1	72.2	393.3	52.1	24.2	274.4	1589.8	55.0	0.0	0.1
[NaphC_1_Im]Tf_2_N (**8**)	586.0	44.7	358.7	1052.8	27.2	340.9	490.8	8.5	0.0	0.3
[(Bz)_2_Im]Tf_2_N (**9**)	547.3	98.6	371.3	46.1	21.2	322.7	368.1	61.3	0.0	0.9
[C_1_Im-C_6_-C_1_Im]Tf_2_N (**10**)	554.8	60.0	379.8	1.70	24.6	312.0	15.7	25.9	0.0	0.1
[C_4_Im-C_6_-C_4_Im]Tf_2_N (**11**)	569.2	57.3	383.7	1.68	20.4	314.3	23.35	28.7	0.0	0.1

## Data Availability

Not applicable.

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
