# Peer review of "A Specific Interaction between Ionic Liquids’ Cations and Reichardt’s Dye"

_molecules, 2022, doi:10.3390/molecules27217205_

Round 1
Reviewer 1 Report
The paper presents computation (mostly) and experimental (some) data for Reichardt’s dye behaviour in several ionic liquids. It is shown that where the positive charge of the cation is delocalized on an aromatic ring (imidazolium), the antibonding orbitals of the positive charged aromatic system are very similar in nature and energy to the LUMO of the Reichardt’s Dye. This led to a specific cation-probe interaction that can be used to elucidate the nature of the ionic liquids’ cation. Parallel computational and experimental investigations have been conducted to elucidate the nature of this interaction with respect to the molecular structure of the cation.
All-in-all, the manuscript presents interesting work and analysis, however, some of the seminal papers published on experimental probing of ionic liquids using Reichardt’s dye in early 2000 are missing from the list of references.
Reviewer 2 Report
The manuscript entitled „A specific interaction between ionic liquids’ cations and Reichardt’s Dye“ written by Angelica Mero, Luca Guglielmero, Lorenzo Guazzelli, Felicia D’Andrea, Andrea Mezzetta, and Christian Silvio Pomelli focusses on computational investigation of both Reichardt`s dye and it’s interaction with various ionic liquid cations as 1 : 1 complexes with the dye as well as UV-Vis spectra of this dye using the same cations. The topic of the MS is interesting. However, the MS needs improvement before publication in the journal molecules.
Synthetic procedures are given for the synthesis of ionic liquids in the experimental part. Unfortunately, no information was found regarding the purity of the synthesized ionic liquids. Content on remaining water in the ionic liquids and melting point or glass transition temperature are missing in the MS.
Furthermore, sample preparation before measurements of the UV-Vis spectra is missing as well in the MS.
Solubility differences of Reichardt’s dye may be observed for ionic liquids containing various anions. Information is missing regarding the solubility of the dye in the ionic liquids under consideration.
From the experimental part and Figure 3 one may assume that ionic liquids containing bromide or bis(trifluoromethylsulfonyl)imide as anion were investigated with Reichardt’s dye. However, the anion was neglected in this MS although the betaine structure of Reichardt’s dye may interact with both the cation and the anion of the ionic liquid.
Furthermore, the ionic liquids consisting of both cation and anion structures may be important for the UV-Vis spectra measured experimentally in this work.
Supplementary Materials mentioned in the MS were also not available.
Figure 1 summarizes various common ionic liquid cations and anions, although only a few of them are in the focus of the MS. The cations under consideration in the MS are summarized in Figure 3. It would be interesting to the reader to get information also regarding the anion of the ionic liquids that were selected for the measurements of the UV-Vis spectra given in Figure 7.
The abbreviation C1C2Mor (Table 1 and Table 2) is not consistent with the abbreviation used for this ionic liquid in Figure 3.
I suggest the MS for publication in the journal molecules after major revision.
Author Response
Reviewer #2:
- Synthetic procedures are given for the synthesis of ionic liquids in the experimental part. Unfortunately, no information was found regarding the purity of the synthesized ionic liquids. Content on remaining water in the ionic liquids and melting point or glass transition temperature are missing in the MS.
Answer, Page 4 Lines 113-116: We thank the reviewer for his/her indications. Prior to the UV-Vis analysis all the tested ionic liquids have been carefully dried, and their water content has been determined by Karl-Fisher titration, observing a concentration lower than 150 ppm for all the tested ionic liquids. We have completed the revised manuscript with this missing information. We understand the concern of the reviewer about the melting point of the tested ionic liquids. As it will be better specified in the following points, only bis-(trifluoromethylsulfonyl)imide ([Tf2N]) ionic liquids have been used for the UV-Vis analyses, while all bromide ionic liquids, mostly solid, were not considered for the study and prepared only as intermediates to obtain the desired [Tf2N] systems. The purity of the bis-(trifluoromethylsulfonyl)imide ([Tf2N]) ionic liquids, after the metathesis reaction (absence of bromide), has been proven by the silver method with fully satisfactory results, as reported in the manuscript. Regarding the tested [Tf2N] ionic liquids, 1,3-bis-(phenylmethyl)imidazolium [Tf2N] ([(Bz)2Im][Tf2N]) and 1-Methyl-3-(naphthalen-2-ylmethyl)-1H-imidazolium [Tf2N] ([Naph-C1Im][Tf2N]) have been found to be low melting solids (m.p. below 50 °C, consistently with the data reported in literature). On the other hand, they were found to remain liquid at room temperature after being molten (behaviour also reported in the literature) for a period long enough to easily allow their use in the UV-Vis tests. All the other tested ionic liquids were found to be liquid at room temperature consistently with the thermal properties reported in literature.
- Furthermore, sample preparation before measurements of the UV-Vis spectra is missing as well in the MS. Solubility differences of Reichardt’s dye may be observed for ionic liquids containing various anions. Information is missing regarding the solubility of the dye in the ionic liquids under consideration.
Answer, Page 4 Lines 109-116: We thank the reviewer for these indications. In the revised manuscript the preparation of UV-Vis samples has been added. The solubility of the Reichardt’s dye in [Tf2N] ionic liquids was found to be adequate for the probe application. On the other hand, no comparisons on the solubility based on the different anion could be done since only [Tf2N] ionic liquids were tested (please refer to the following answers). As said in the caption of Figure 7 the spectra has been re-normalized and the vertical axis is in arbitrary units.
- From the experimental part and Figure 3 one may assume that ionic liquids containing bromide or bis-(trifluoromethylsulfonyl)imide as anion were investigated with Reichardt’s dye. However, the anion was neglected in this MS although the betaine structure of Reichardt’s dye may interact with both the cation and the anion of the ionic liquid. Furthermore, the ionic liquids consisting of both cation and anion structures may be important for the UV-Vis spectra measured experimentally in this work.
Answer, Page 6 Lines 240-243, Page 7 Lines 266-267: We agree with the reviewer that this point has not been properly clarified in the original version of the manuscript. UV-Vis spectra have been recorded only for ionic liquids featuring a bis-(trifluoromethylsulfonyl)imide anion. The missing indications about the type of anion for the tested ILs has now been added in the revised manuscript. We agree with the reviewer that the anions should also be considered for their interactions with the Reichardt’s dye, anyway the conducted calculations showed that the interactions of the bis-(trifluoromethylsulfonyl)imide anion can be considered negligible. This point has been clarified in the revised version of the manuscript.
- Supplementary Materials mentioned in the MS were also not available.
We apologies to the reviewer for this inconvenience. The Supplementary Materials has been regularly uploaded in the submission process.
- Figure 1 summarizes various common ionic liquid cations and anions, although only a few of them are in the focus of the MS. The cations under consideration in the MS are summarized in Figure 3. It would be interesting to the reader to get information also regarding the anion of the ionic liquids that were selected for the measurements of the UV-Vis spectra given in Figure 7.
Answer, Page 7 Figure 3, Page 7 lines 266-267, Page 8 Table 1, Page 11 Table 2, Page 13 Figure 7 and Table 3: We agree with the reviewer about the need of providing a more evident indication about the structures of the ionic liquids selected for the UV-Vis measurements. The different cations are all paired with the bis-(trifluoromethylsulfonyl)imide anion. We made this information clear in the amended version of the manuscript. A detailed explanation about the choice of the anion has been also added.
- The abbreviation C1C2Mor (Table 1 and Table 2) is not consistent with the abbreviation used for this ionic liquid in Figure 3 and Figure 7.
Answer, Page 7 Figure 3, Page 13 Figure 7: We thank the reviewer for highlighting this mistake which has been corrected it in the revised manuscript.